# Multilingual Automatic Speech Recognition for Scandinavian Languages

**Rafal Cerniavski**
Uppsala University
Conversy AB
rafal.cerniawski@gmail.com

**Sara Stymne**
Department of Linguistics and Philology
Uppsala University
sara.stymne@lingfil.uu.se

## Abstract

We investigate the effectiveness of multilingual automatic speech recognition models for Scandinavian languages by further fine-tuning a Swedish model on Swedish, Danish, and Norwegian. We first explore zero-shot models, which perform poorly across the three languages. However, we show that a multilingual model based on a strong Swedish model, further fine-tuned on all three languages, performs well for Norwegian and Danish, with a relatively low decrease in the performance for Swedish. With a language classification module, we improve the performance of the multilingual model even further.

## 1 Introduction

Automatic speech recognition (ASR) is the task of transforming speech into text, often referred to as transcription. Multilingual ASR tackles the task in multiple languages with the same model or pipeline. Modern ASR architectures such as DeepSpeech (Hannun et al., 2014), Wav2Vec (Baevski et al., 2020), and Whisper (Radford et al., 2022) are capable of transcribing speech with Word Error Rates (WERs) well below 10 percent. To achieve this, models require copious amounts of data, which is unavailable for the vast majority of languages. For low-resource languages, multilingual models as means of bootstrapping the performance are often the only solution.

Conneau et al. (2021) demonstrate that a multilingual setting can be beneficial even for high-resource languages. Pratap et al. (2020), however, suggest that limiting models to smaller, typologically related languages is more productive than training on all languages at once. As such, it can be argued that Scandinavian languages are a great fit for multilingual NLP models. Swedish, Danish,

and Norwegian all originate from old Norse and share numerous similarities, such as largely overlapping lexicons and similar grammar. As noted by Delsing and Lundin Åkesson (2005), the similarities across the three languages are not linear, since Swedish and Norwegian are most similar in speech, whereas Danish and Norwegian are most similar in writing. Nevertheless, Sahlgren et al. (2021) argue that Scandinavian languages are so similar that large text-based language models for these languages should be trained jointly. It has also been shown that utilizing the similarities between the Scandinavian languages can improve text-based tasks such as machine translation (e.g. Tiedemann, 2009) and parsing (e.g. Smith et al., 2018). However, to the best of our knowledge, there is no work where the usefulness of combining the Scandinavian languages is reported for speech-based tasks, such as ASR.

We focus on identifying whether a multilingual ASR model for Swedish, Danish, and Norwegian can be trained to utilize an existing high-quality monolingual model, as we fine-tune a strong Swedish end-to-end model to also handle the Danish and Norwegian languages. In addition, we analyze how well the monolingual ASR models transfer across the Scandinavian languages in a zero-shot setting. We also evaluate how the multilingual setting affects the quality of transcription as opposed to monolingual settings. Lastly, we show that a language classification module can be used for selecting a language model in the multilingual setting. We conduct all experiments on the Wav2Vec 2.0 (Baevski et al., 2020) based ASR models. Additional experiments, as well as more in-depth analysis, can be found in Černiavski (2022).

## 2 Previous Work

**Language Models for ASR** The usage of a language model in speech recognition has contin-

uously proven to boost the quality of transcription. Positive results have been observed with both statistical n-gram language models (Amodei et al., 2016; Håkansson and Hoogendijk, 2020) and transformer-based models, such as BERT (Baevski et al., 2020). Most considerable improvements seem to result from domain-specific language models, as contextualization and biasing of models have repeatedly improved the quality of transcription (Aleksic et al., 2015).

**Multilingual ASR** Transcription of multiple languages via a single model or pipeline has been made possible through a variety of architectures. Approaches range from an assemble of monolingual models connected through a preceding language classification component (Lyu and Lyu, 2008; Mabokela and Manamela, 2013; Barroso et al., 2010), to models sharing the phone models (Lin et al., 2009) or hidden layers of acoustic models (Yu and Deng, 2015) across two or more languages, to being conjunct on all levels, sharing all components and treating all input languages as one (Pratap et al., 2020; Conneau et al., 2021).

As a general rule, the effects of a multilingual setting on the quality of transcription are twofold. Low-resource languages tend to reap the benefits, as models seemingly generalize from the patterns learned in higher-resource languages (Yu and Deng, 2015; Bhable and Kayte, 2020). High-resource languages, however, tend to suffer (Lin et al., 2009; Conneau et al., 2021), likely due to the noise introduced through the exposure of models to data in (a) foreign language(s). Nevertheless, Pratap et al. (2020) demonstrated that there appear to be ways of mitigating the toll of a multilingual setting on the resource-rich languages by means of a typologically motivated choice of languages in a cluster as well as cluster-specific rather than one-for-all decoders.

## 3 Methodology

We first evaluate the performance of monolingual Swedish, Danish, and Norwegian models on the test sets of each language (i.e. the Swedish model was evaluated on Swedish, Danish, and Norwegian test sets). For comparison, we also evaluate the performance of an English ASR model on the three Scandinavian languages. We do so first to obtain comparable word error rates of each model for their intended language, except for English; second, to explore a zero-shot setting, where we explore whether the typological similarity of Scandinavian languages enables the ASR models trained on one of the languages to transcribe the data in the other two languages. We add English, a more distant Germanic language, for comparison.

In a second set of experiments, we fine-tune trial multilingual ASR models for Swedish, Danish, and Norwegian. We aim to utilize the high quality of the already fine-tuned Swedish model (Malmsten et al., 2022) to bootstrap the transcription in Danish and Norwegian as opposed to training a model on the three languages from scratch. As such, we attempt fine-tuning the Swedish model in the following settings:

1. **Retraining DA+NO** - using complete training sets in Danish and Norwegian, with no Swedish training data (30,000 entries total)

2. **Retraining DA+NO+SE_half** - using complete training sets in Danish and Norwegian, and half of the Swedish training data (37,500 entries in total)

3. **Retraining DA+NO+SE_full** - using complete training sets of all three languages (45,000 entries in total)

For comparison, we also train a model on all three languages (15,000 entries per language, 45,000 total) on the pre-trained, but not fine-tuned Swedish model[1]. We train these models for 5 epochs and evaluate on the trilingual development set every 1,000 updates. In order to investigate the effect of adding a language model for the multilingual models, we train a language classifier, see Section 5 for details.

In our final experiment, we select the best-performing trial model (**Retraining DA+NO+SE_full**) to train a multilingual model for 20 epochs. We evaluate the model in two settings. In the first, we use no language model in the decoding. In the second, we use the language classifier to predict the language, in order to select a 4-gram language model for the predicted language. We train the 4-gram language models on the entirety of the original NST training sets, except for Swedish, where we exclude the sample used as test set.

We report Word Error Rate as our main evaluation metric and perform a brief qualitative analysis of the most common errors.

---

[1] https://huggingface.co/KBLab/wav2vec2-large-voxrex

## 4 Data and Models

**Data** We created testing, training, and development subsets for Swedish, Danish, and Norwegian from two datasets: Nordisk Språkteknologi (NST)[2] and Common Voice (CV) 8.0 (Ardila et al., 2020). For testing subsets of Danish and Norwegian, we used the entire test sets from NST, which amount to 77.1 and 115.3 hours of speech respectively. For Swedish, due to the lack of a modernized version of the NST test at the time of working on the project, we randomly sampled 20% of the training set - roughly 73,2 hours of speech. For training, we limit the subsets to 15,000 entries per language (roughly 7 hours of speech) per language due to limited computational resources. We ensure that the Swedish train and test subsets do not overlap.

We use the CV dataset to construct a validation set for the multilingual models. For Swedish and Danish, we randomly sampled 2,000 validated entries per language. No validated data were available for Norwegian Bokmål; we, therefore, used a held-out sample of 2,000 entries from the Norwegian NST training dataset. In total, we used roughly one hour of speech per language for validation.

We processed each subset by downsampling the audio to 16 kHz and normalizing the transcriptions. The normalization involved lower-casing all characters and removing non-alphanumeric characters, such as punctuation markers.

**Models** For monolingual baselines in Swedish, Norwegian, and English, we used Wav2Vec 2.0 models publicly available on Huggingface[3]. Due to the lack of an existing fine-tuned Danish model at the time, we fine-tuned one ourselves: we used the publicly available pre-trained Danish Wav2Vec 2.0 model[4], which we then fine-tuned on one GPU for 10 epochs on our Danish NST train subset. In the encoder, we retain the original pa-

---

[2]Swedish: `https://www.nb.no/sprakbanken/en/resource-catalogue/oai-nb-no-sbr-56/`;
Danish: `https://www.nb.no/sprakbanken/en/resource-catalogue/oai-nb-no-sbr-55/`;
Norwegian: `https://www.nb.no/sprakbanken/en/resource-catalogue/oai-nb-no-sbr-54/`
[3]Swedish: `https://huggingface.co/KBLab/wav2vec2-large-voxrex-swedish`
Norwegian: `https://huggingface.co/NbAiLab/nb-wav2vec2-1b-bokmaal`
English: `https://huggingface.co/facebook/wav2vec2-base-960h`
[4]`https://huggingface.co/Alvenir/wav2vec2-base-da`

| Model | Test Set | No LM | With LM |
|---|---|---|---|
| Swedish | Swedish | 2.19% | 2.74% |
| | Danish | 78.58% | 72.69% |
| | Norwegian | 61.78% | 52.06% |
| Danish | Swedish | 120.10% | 98.93% |
| | Danish | 19.14% | 13.82% |
| | Norwegian | 104.56% | 90.06% |
| Norwegian | Swedish | 83.51% | 73.82% |
| | Danish | 83.79% | 75.05% |
| | Norwegian | 16.47% | 12.03% |
| English | Swedish | 110.06% | 93.59% |
| | Danish | 99.50% | 88.71% |
| | Norwegian | 102.52% | 90.35% |

Table 1: WERs of monolingual models on the three Scandinavian languages, no language model versus a 4-gram language model.

rameters of the pre-trained model, whereas in the decoder, we set the batch size to 10, gradient accumulation steps to 3, learning rate to 1e-4, and weight decay to 0.005.

## 5 Language Classification Module

The language classification module is initialized on top of the same pre-trained Swedish Wav2Vec 2.0 model we used for ASR. We train it on 15,000 entries per language, randomly sampled from the train sets (45,000 entries in total). We set the batch size to 4, learning rate to 1e-4, and gradient accumulation steps to 2, and use mean pooling.

We evaluate the classification module on a concatenation of the test sets from all three languages, with results in Figure 1. The classifier reached an overall accuracy of 98% across the three languages, with very few confusions between Danish and Swedish. It is also noticeable that most errors occur for short segments, often containing a single word. For segments of at least five seconds, the accuracy is near perfect.

## 6 Results and Discussion

WERs for Swedish, Danish, Norwegian, and English ASR models on the three Scandinavian languages are shown in Table 1. The results on zero-shot ASR are poor. We can see some general patterns in the performance across languages. The Swedish and Norwegian models perform better for all three Scandinavian languages than the English model. However, the Danish model performs as poorly on Swedish and Norwegian data as the En-

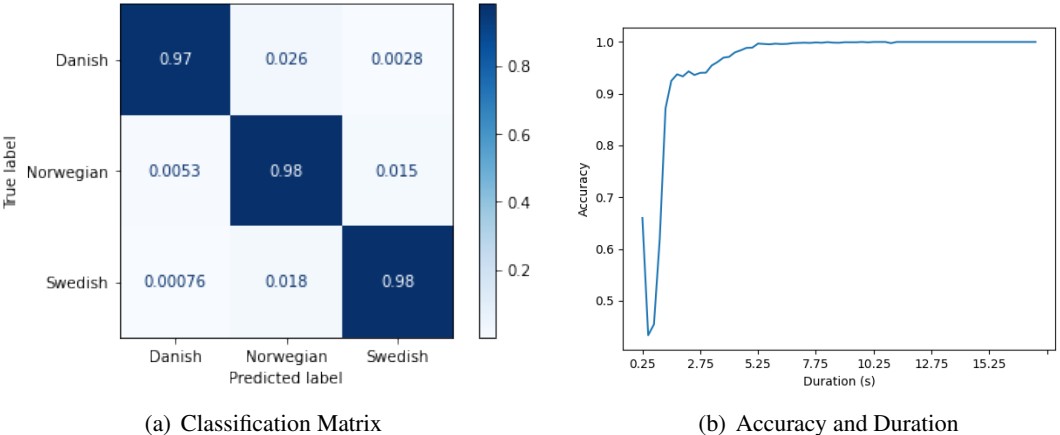

(a) Classification Matrix        (b) Accuracy and Duration

Figure 1: Evaluation of the language classification module. The accuracy in subgraph b is averaged over the three languages.

glish model. This could stem from the fact that our Danish model was trained on very little data compared to the Swedish and Norwegian models, as the Danish model performs poorly even on the Danish data despite being trained on in-domain data. However, it could be affected by the fact that the pronunciation in Danish is quite different from Swedish and Norwegian.

Even though the results are poor, we note that the results for the Scandinavian languages largely follow the patterns for mutual intelligibility between human speakers (Delsing and Lundin Åkesson, 2005); the Swedish ASR model is better at transcribing Norwegian than Danish, the Danish model is better for Norwegian than Swedish, and the Norwegian model is somewhat better for Swedish than Danish. The latter difference is more pronounced for character error rates, see (Černiavski, 2022).

The scores confirm that the addition of a simple n-gram language model leads to stable improvements of the quality in transcription, even in a cross-lingual setting. The Swedish model is an exception, though, likely due to the overall high quality of the model, which is only limited by such a language model.

Lastly, qualitative analysis of the outputs reveals that some of the predictions considered to be errors due to a deviation from the ground truth are grammatically correct alternative spellings that can have the same pronunciation. For instance, in the output of the monolingual Swedish model, some of the most common substitution errors are *skall* instead of *ska*, *i stället* instead of *istället*, and

*i dag* instead of *idag*. Due to the usage of WER as an evaluation metric, the latter two examples are treated as 2 errors each. This is because WER considers *istället* to be substituted with *stället* and treats the preposition *i* to be an insertion error. Similar patterns can be observed in the outputs for the other two languages, which leads us to believe that WER might not be the most suitable evaluation metric for Scandinavian, and possibly other, languages.

WER for the trial multilingual models are shown in Figure 2. The results indicate that the initialization of the multilingual model from a monolingual model is only effective in low-resource settings. This is because a model trained from scratch on all three languages reaches comparable WER within roughly 5,000 steps. Nevertheless, despite the subtle difference, the average WERs on all three languages indicate that the model initialized from a fine-tuned Swedish model and further fine-tuned on complete training sets (**Retraining DA+NO+SE_full**) is second only to monolingual baselines. Analogous patterns can be seen in terms of character error rates (Černiavski, 2022). Hence, we choose this setting for training our final multilingual model.

The scores of the final multilingual ASR model able to transcribe Swedish, Danish, and Norwegian, as opposed to the monolingual baselines are shown in Table 2. Using a language classification model for selecting which language model to add, leads to improvements for all three languages. We observe stable improvement over monolingual baselines for Norwegian and Danish both with

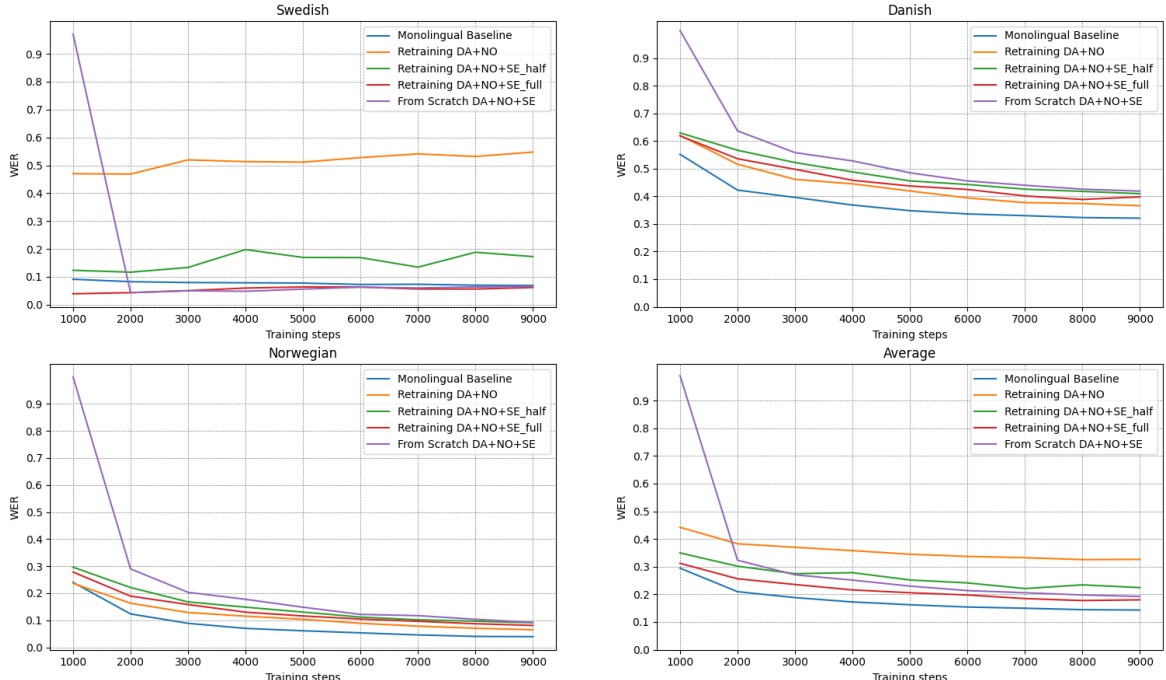

Figure 2: Word Error Rates of the multilingual trial models and a monolingual baseline on the evaluation set, mapped over training steps.

and without language models, with only a slight drop in performance for Swedish. However, more analysis is needed to investigate the influence of matching the language versus matching the domain since both our training and test sets are from the NST dataset.

We observe that the multilingual model performs significantly better in Norwegian than it does in Danish, which can also be seen from the progression in the WERs of the trial models shown in Figure 2. This is likely because the development and test sets for Norwegian we used were from the same domain, which was not the case for Danish, but it may also be influenced by Norweigan pronunciation being closer to Swedish than Danish to Swedish. Černiavski (2022) presents a more detailed qualitative analysis of the transcription in the monolingual versus multilingual setting, as well as with and without LM settings. We observe that cross-lingual errors (e.g. when a Swedish word is transcribed with a Norwegian spelling) are very rare in a multilingual setting even when LMs are not used.

## 7 Conclusions

Multilingual automatic speech recognition is often considered to be useful only for low-resource

| Test Set | Model | No LM | With LM |
|---|---|---|---|
| Swedish | Mono | **2.19%** | 2.74% |
| | Multi | 4.61% | 3.26% |
| Danish | Mono | 19.14% | 13.82% |
| | Multi | 12.69% | **10.43%** |
| Norwegian | Mono | 16.47% | 12.03% |
| | Multi | 9.64% | **6.51%** |

Table 2: The performance **Mono**lingual baselines versus our **Multi**lingual model.

languages. Though a multilingual model can hardly compete in ultra-high-resource languages, we show that the multilingual Scandinavian model can perform comparably or even perform better than monolingual models. Our results indicate that it could be useful to combine the Scandinavian languages not only for text, but also for speech processing. More extensive evaluation of models is needed to conclude whether the model benefits from a multilingual setting, or only from in-domain training. Further research could also explore the effects of a multilingual setting on the ability to classify dialects of Scandinavian languages.

## Acknowledgments

Computations were enabled by resources in project UPPMAX 2020/2-2 at the Uppsala Multidisciplinary Center for Advanced Computational Science.

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
