# OpenReview forum: "Multilingual Automatic Speech Recognition for Scandinavian Languages"
_NoDaLiDa/2023/Conference — NoDaLiDa 2023_

### Official Review · Reviewer_nYDh · 2023-03-08
**More data helps sometimes**

**Rating:** 7
**Confidence:** 4

**Review:**

This paper reports on experiments where a monolingual model is fine-tuned with data from other Scandinavian languages

I am a bit confused about the exact experimental setup and I suggest you make sure the naming conventions match. E.g. I don't see any evaluation for swedish wav2vec model fine-tuned on Danish and Norwegian.

The results confirm what we have observed for other languages in terms of data augmentation: if you do not have enough relevant data to estimate a good monoligual model, using other data can help, but if you have enough data to estimate the parameters of your model, then using perturbation or other data augmentation techniques becomes unhelpful in terms of performance.

The content is good for a short paper, but even for a short paper, I would expect some kind of error analysis: are there any patterns to what is improved by adding mulitlingual data? are there patterns to the decrease in performance on Swedish? E.g. when we use a multilingual model like XLS-R and fine-tune for Danish, performance on names that are not originally Danish improves a lot. And in that vein, compare to to fine-tuning on a model with other languages than scandinavian would be relevant.

Finally, I recommend that you use the CV data as an extra test set and subdivide the large test sets into test and validation instead.

**Paper Type:**

Short paper

---

### Official Review · Reviewer_drMP · 2023-03-09
**Multilingual Automatic Speech Recognition for Scandinavian Languages**

**Rating:** 6
**Confidence:** 5

**Review:**

The paper describes experiments on various combinations of training/fine-tuning ASR models for the Scandinavian languages, both in matched language and mismatched language conditions.

Some of the observations are obvious. E.g. that applying a language-specific model to a different language than it is trained for will result in poor (or very poor) performance. Although it is tempting to accept the observation that the performance in these cases will be related to the linguistic difference between the languages, performance in the WER range > 80% (slightly less with language models) imply too many errors to discount that the observations may actually be caused by random variations.

The language is ok, although there are some typos, but the presentation unfortunately has several flaws, in particular lacking important details. It would, e.g., be of interest to include what type of architecture is used in the models, as well as some information about the complexity of the models.

Wav2vec2 and Whisper are included in a list of monolingual ASR architectures in the Introduction. While they can be trained on a single language, both these architectures can be, and have been, trained on multilingual data. Wav2vec2, it should be noted, though, typically requires language specific data for fine-tuning the decoder for speech transcription. However, the amount of language specific data can be very modest (~10 hours).

>For Swedish, due to the lack of a modernized version of the NST test set, we randomly sampled…

What does this mean? The NST data for all three languages has been reorganised by the Norwegian Språkbanken and are freely available. Nothing special about the Swedish data - the collection was all made in the same period.

The use of a language classifier to select the appropriate, language specific language model is reported in Table 2. In appendix the performance of the classifier is reported as an accuracy of ~98% for all languages. This seems extraordinary good, even for a 3-class task, and it would have been useful to have a more detailed description of the classfier architecture – there is in fact no description of the classifier, only some information on the training procedure.
The statement in the paper that
>It gives an overall strong performance of at least 0.97 percent for each language pair

Is of course a typo, it should read ‘…at least 97%...’

>For comparison, we also fine-tune a model on all three languages (15,000 entries per language) from scratch on the pre-trained (but not fine-tuned) Swedish model.

To fine-tune from scratch is a contradiction in terms. Presumably, this means that the encoder part is kept while the decoder is re-initialized and trained using the same architecture as the pre-trained Swedish model?

Regarding the mono-lingual Danish model:
>Due to the lack of an existing fine-tuned Danish model, we fine-tuned on ourselves: we used the publicly available pre-trained Danish Wav2Vec model, which we then fine-tuned on one GPU for 10 epochs on our Danish NST train subset

I.e. on the same domain as the test data - different from the Swedish and Norwegian models which are fine-tuned on VoxRex (probably) and NPSC respectively, which one could suspect gave an edge to the Danish matched language performance - which is not evident from the results.

It appears that the CommonVoice data is only used as a validation set during training. It should be noted that the NST and CV data have significant differences both in terms of linguistic content and recording conditions. Since evaluation is done on the NST data, using CV for validation might not lead to the best results. Since CV data did not exist for Norwegian Bokmål, NST data (i.e. in-domain) were used for validation - did you observe any differences that could be attributed to these issues?

In the Conclusion it is stated that
>Multilingual automatic speech recognition is often considered to be useful only for low-resource languages

I believe this is a sentiment that has been defeated by the performance by Whisper and other high-performance multilingual systems that can provide fairly competitive performance also for high-resource languages - but require massive amounts of training data.

**Paper Type:**

Short paper

---

### Official Review · Reviewer_x3se · 2023-03-12
**Very interesting paper on how to utilize a combination of a language family-based model for ASR and a language-specific LM to gain improvements even over fully monolingual models in some cases.**

**Rating:** 8
**Confidence:** 3

**Review:**

The paper is of high quality and clearly written. The evaluation is timely and provides new insights into how medium-sized training data can be abrigded with multilingual data. This adds significantly to the methodology we need for how to deal with under-resources languages.

The pros of the paper is that it shows the results on languages that are not traditionally considered low-resourced and demonstrates that improvements can still be achieved.

The cons of the paper is that it could have been a long paper with more details and error analysis, and I would eagerly have read more about how this methodology can be used for recognizing dialectal or colloquial varieties of the same languages.

**Paper Type:**

Short paper

---

### Decision · Program_Chairs · 2023-03-17

Accept